# Defining New Pathways to Manage the Ongoing Emergence of Bat Rabies in Latin America

**DOI:** 10.3390/v12091002

**Published:** 2020-09-08

**Authors:** Julio A. Benavides, William Valderrama, Sergio Recuenco, Wilson Uieda, Gerardo Suzán, Rafael Avila-Flores, Andres Velasco-Villa, Marilene Almeida, Fernanda A.G. de Andrade, Baldomero Molina-Flores, Marco Antonio Natal Vigilato, Julio Cesar Augusto Pompei, Paolo Tizzani, Jorge E. Carrera, Darcy Ibanez, Daniel G. Streicker

**Affiliations:** 1Departamento de Ecología y Biodiversidad, Facultad de Ciencias de la Vida, Universidad Andrés Bello, Santiago 8370146, Chile; 2Institute of Biodiversity, Animal Health and Comparative Medicine, College of Medical Veterinary and Life Sciences, University of Glasgow, Graham Kerr Building, Glasgow G12 8QQ, UK; 3Centro de Investigación para la Sustentabilidad, Facultad de Ciencias de la Vida, Universidad Andrés Bello, 8370146 Santiago, Chile; 4Association for the Conservation and Development of Natural Resources ILLARIY, Lima 051, Peru; wvalderrama@illariy.org; 5Departamento de Pediatría, Obstetricia y Ginecología y de Medicina Preventiva, Universidad Autónoma de Barcelona, 08007 Barcelona, Spain; 6Facultad de Medicina San Fernando, Universidad Nacional Mayor de San Marcos, Lima 15001, Peru; sergio.recuenco@gmail.com; 7Departamento de Zoologia, Universidade Estadual Paulista Júlio de Mesquita Filho Campus de Botucatu, Botucatu 18618-970, Brazil; wuieda@hotmail.com; 8Laboratorio de Ecología de Enfermedades y Una Salud, Facultad de Medicina Veterinaria y Zootecnia, Universidad Nacional Autónoma de México, México City 04510, Mexico; gerardosuz@gmail.com; 9División Académica de Ciencias Biológicas, Universidad Juárez Autónoma de Tabasco, Villahermosa 86150, Mexico; rafaelavilaf@yahoo.com.mx; 10Division of High-Consequence Pathogens and Pathology, National Center for Emerging and Zoonotic Infectious Diseases, Centers for Disease Control and Prevention, 1600 Clifton Rd, NE, Atlanta, GA 30329, USA; dly3@cdc.gov; 11Centro de Controle de Zoonoses da Prefeitura do Município de São Paulo, São Paulo 02031-020, Brazil; lenefalmeida@hotmail.com; 12Departamento de Biologia, Instituto Federal de Educação, Ciências e Tecnología do Pará, Tucuruí 68-455-695, Brazil; fernanda.atanaena@ifpa.edu.br; 13Pan-American Center for Foot-and-mouth Disease and Veterinary Public Health—Pan American Health Organization/World Health Organization (PANAFTOSA—PAHO/WHO), Rio de Janeiro 25045-002, Brazil; molinab@paho.org (B.M.-F.); vigilato@paho.org (M.A.N.V.); jcpompei@paho.org (J.C.A.P.); 14OIE-World Organisation for Animal Health, 75017 Paris, France; p.tizzani@oie.int; 15Departamento de Mastozoología, Museo de Historia Natural, Universidad Nacional Mayor de San Marcos, Lima 15072, Peru; jecarrerag@gmail.com; 16Programa de Conservación de Murciélagos de Perú, Lima 15072, Peru; 17Director Regional de la Dirección Regional Sectorial Agraria, Gobierno Regional de Apurímac, Abancay 03001, Peru; darcymu@gmail.com; 18MRC–University of Glasgow Centre for Virus Research, Glasgow G61 1QH, UK

**Keywords:** rabies lyssavirus, zoonotic disease, control measures, public health, cross-species transmission

## Abstract

Rabies transmitted by common vampire bats (*Desmodus rotundus*) has been known since the early 1900s but continues to expand geographically and in the range of species and environments affected. In this review, we present current knowledge of the epidemiology and management of rabies in *D. rotundus* and argue that it can be reasonably considered an emerging public health threat. We identify knowledge gaps related to the landscape determinants of the bat reservoir, reduction in bites on humans and livestock, and social barriers to prevention. We discuss how new technologies including autonomously-spreading vaccines and reproductive suppressants targeting bats might manage both rabies and undesirable growth of *D. rotundus* populations. Finally, we highlight widespread under-reporting of human and animal mortality and the scarcity of studies that quantify the efficacy of control measures such as bat culling. Collaborations between researchers and managers will be crucial to implement the next generation of rabies management in Latin America.

## 1. Introduction

Rabies is among the oldest known zoonoses but still challenges public and animal health systems on most continents. In Latin America, national and regional programs focusing on mass dog vaccination have left only lingering hotspots of canine-mediated rabies in a few countries [1]. However, rabies transmitted by *Desmodus rotundus* (*Desmodus rotundus* rabies virus, DRRV), generally referred to as the common vampire bat, has emerged as a previously underappreciated and growing threat which now causes the majority of human and livestock rabies mortality in countries of Latin America where canine rabies is in the verge of elimination [1,2,3]. Although the human health burden of DRRV is challenging to measure given geographic isolation of affected communities and under-reporting, rates of bat depredation on humans and the associated rabies mortality can be alarming. Surveys indicate that 23–88% of inhabitants in high-risk areas are bitten by *D. rotundus*, leading to mortality of 1–7% of local human populations (1–39 deaths per village) when outbreaks occur (summarized in [4]). Agricultural losses are also substantial. Tens of thousands of livestock die from DRRV annually, costing over $30 million USD before considering under-reporting, recurrent investments in surveillance, diagnostics and prevention [2,5]. The impacts of DRRV led to the establishment of national control and prevention programs across Latin America. Activities include surveillance, culling bats, vaccination of livestock, and pre- and post-exposure prophylaxis of humans [6]. However, limited funding cannot cover all populations at risk. More worrying, bat bites in humans were historically associated with remote rural communities, but now also occur in urban areas and in coastal fishing communities [7,8]. New evidence that *D. rotundus* feeds on and transmits rabies to domestic dogs may complicate international efforts to eliminate canine-mediated rabies [9]. The geographic area impacted is also increasing, with spatial expansions of either *D. rotundus* or DRRV observed in Mexico, Peru and Uruguay, speculated in some cases to arise from climate change [6,10,11,12]. Even within the core affected range, multiple countries experienced increases in reported DRRV over the past decade, most notably in Central America and western South America (Figure 1). 

Motivated by the recognition of DRRV as a historically persistent and emerging problem in Latin America, this review presents a consensus from individuals representing academic, government and international health organizations on the current knowledge of the epidemiology of DRRV in its natural reservoir and the factors that contribute to spillover to humans and livestock. Beyond rabies, thousands of animals and humans are bitten and fed upon by *D. rotundus* every night, creating a transmission route for diverse viral, bacterial and parasitic pathogens (including secondary infections in bite wounds), and causing additional economic losses to ranchers associated with anemia from blood loss in production animals [13,14,15,16]. We, therefore, simultaneously address the issue of bat bites. We concentrate on four key pre-requisites for cross-species transmission: reservoir host distribution, transmission dynamics within the reservoir host, drivers of cross-species exposures, and recipient host susceptibility [17]. For each component, we discuss the management tools currently available and identify research gaps that must be filled for current and future DRRV control strategies to be effective (Figure 2). 

## 2. Reservoir Host Distribution

### 2.1. Current Knowledge

Three extant species of hematophagous bats are the common vampire bat (*Desmodus rotundus*), the hairy-legged vampire bat (*Diphylla ecaudata*), and the white-winged vampire bat (*Diaemus youngi*). While the other two species feed mainly on birds and wildlife [13], *D. rotundus* regularly feeds on livestock and occasionally on humans, and is the sole reservoir of DRRV [13,18,19]. Understanding what drives the geographic distribution of *D. rotundus* is essential to understand the risk of DRRV. *D. rotundus* occurs throughout tropical and subtropical Latin America, from northern Mexico to northern Argentina and Chile [11], and occupies different ecosystems including tropical rainforests, coastal deserts, xeric shrublands, and mountainous regions up to 3600 m [11]. Factors influencing bat presence and rabies risk at large spatial scales include temperature, altitude and precipitation, whereas, at local scales, livestock density, human-induced forest fragmentation, density/proximity of highways and rivers appear to be important [11,20,21]. Growth in the *D. rotundus* population over the last century relates to an increase in access to livestock as a food source and availability of man-made structures that could be used as roosts including mines, tunnels, wells, culverts, and abandoned houses [6,21]. In contrast, large-scale deforestation of trees used by *D. rotundus* for roosting may reduce their relative abundance [6,22]. Avoidance of open areas as flight routes may also explain the close association of *D. rotundus* presence with forest fragments, rivers and tree lines [23].

### 2.2. Current Management Practices

Control measures to reduce the geographic distribution and population size of *D. rotundus* have been applied for half a century [24]. National campaigns use orally ingested anticoagulant poisons (“vampiricide”) that are applied topically and spread among bats by allogrooming, or are applied to livestock and ingested when bats feed [6]. Although potential exists for ingestion of anticoagulant poisons by other bat species or through contamination of shared roosts [25], *D. rotundus* has been observed to occupy specific sites within roosts, which may reduce impacts on other species. Unauthorized methods for bat population control include capture and killing of bats using nets or improvised traps at roosts as well as destruction, damaging, sealing or burning of roosts [26]. These activities have caused widespread mortality among vampire bats and other bat species, raising ethical concerns among conservationists as well as reducing the ecosystem services provided by bats [26,27]. They should be actively discouraged by governments. Irrespective of the methods used, culling temporarily reduces *D. rotundus* populations and alleviates bites on humans and/or livestock at local scales [28,29], but is unlikely to alter the geographic distribution of *D. rotundus* at larger scales unless the frequency and intensity of campaigns is increased to unprecedented levels or novel technologies for bat population control are developed.

### 2.3. Improving Management

Existing management focuses exclusively on increasing bat mortality. Landscape-level interventions could target other determinants of *D. rotundus* presence and abundance such as the availability of human-provided food and shelter (Figure 2). Ecological interventions that increase the presence of predators (e.g., domestic house cats) have been proposed [30], but should be considered with caution given the possible effects of cats on native wildlife and their ability to transmit DRRV to people. Reducing the availability of livestock prey is another option, but risks encouraging bat dispersal which could increase rabies spread or increasing bat depredation on alternative prey like humans. An exciting prospect involves developing hormonal reproductive control technologies as used in wild terrestrial mammals [31,32]. Relative to culling, reproductive suppression might more gradually reduce the size of *D. rotundus* populations (minimizing the social disturbance which has been hypothesized to heighten rabies spread [33]), while skewing the age structure towards adults that may have immunity against rabies from prior exposures. Ecological models may also provide new opportunities to increase the efficiency of interventions by identifying areas where bat populations may be particularly sensitive or robust to intervention. Finding the optimal balance among emerging choices may also face location-specific constraints in what is practically achievable and effective [34].

## 3. Transmission Dynamics within the Reservoir

### 3.1. Current Knowledge

DRRV is believed to primarily be transmitted between bats through bites. As typical for a rabies virus, all mammalian species are susceptible and successful infections invariably result in acute and lethal encephalitis. Long-term maintenance occurs through a species-specific transmission cycle within bats [12,35]. In bats, non-lethal infections, where individuals clear infection prior to becoming infectious, are regularly observed in the wild and in captive infection studies via the presence of virus-neutralizing antibodies (VNAs) [33,36,37]. The protective nature of naturally acquired VNAs remains unclear. The prevalence of active infection in free living bats is generally low (~1%) [38]. However observations of wild bats brought into captivity have shown higher proportions of infected bats shedding virus on their saliva, although stress related to captivity could have increased their susceptibility to the virus [39]. Although undersampling of active infections cannot be discarded, one likely explanation for this apparent discrepancy is that the virus undergoes localized epizootics, during which prevalence is substantially elevated, followed by local extinctions. Indeed, such invasion–extinction dynamics are readily observed in the complex spatiotemporal patterns of spillover to livestock, including epizootic waves, where DRRV spreads across the landscape via transmission between neighboring bat colonies at a relatively constant speed [10] and metapopulation persistence, where asynchronous presence of the virus in bat colonies prevents extinction at a larger spatial scale, and localized lineage replacements [33,40,41]. Extended incubation periods can occur but are a less plausible explanation for localized disappearances of DRRV strains over multiple years suggested by passive surveillance of livestock [40]. Confirming disappearance of DRRV from bats themselves has so far been hampered by the short time frame of most longitudinal studies and their reliance on antibody-based diagnostics (which remain detectable for months after viral clearance) rather than direct evidence of viral circulation by RT-PCR [36,42]. Assuming extinction–recolonization dynamics are confirmed, movement of infected bats between colonies is, therefore, crucial for long-term viral maintenance [33]. The mechanisms of viral dispersal are beginning to be understood. Although *D. rotundus* is non-migratory and typically has small home ranges with common dispersals over relatively short distances (1-3 km), longer distance dispersals (up to 54 km) have been documented [43,44]. Recent molecular studies suggested male bat dispersal enables DRRV spread between colonies [40,43], highlighting males as a potentially important target for management.

### 3.2. Current Management Practices

The only practice currently employed to reduce DRRV transmission within the reservoir is the reduction in bat density through culling. In theory, culling of reservoirs for disease control is most effective when the pathogen transmission rates depend on host density, with a critical density threshold under which the pathogen cannot be maintained by the population [45]. In practice, the relationship between reducing bat populations and rabies risk is complex [28,36]. The social disruption of culling bats might facilitate rabies spread by increasing bat dispersal [33]. Further complications could arise if culls reduce population immunity by preferential killing adult immune individuals or if vacated niche space increases juvenile survival or immigration of naïve individuals [46]. While empirical data remain limited, a comparison of seroprevalence between *D. rotundus* colonies with different culling histories suggested that culling was associated with higher rather than lower seroprevalence in bats [36]. Despite national programs of bat culling running for several decades, there is little published evidence that culling reduces DRRV transmission to humans or cattle. In particular, it is unclear whether culling programs are ineffective because of reactive culling (i.e., after a DRRV is detected on humans or livestock) or a lack of resources to sustain culling efforts over time, partly driven by the lack of an evident base identifying what levels of culling might be effective. In Argentina, experimental gassing of roosts with cyanide reduced *D. rotundus* populations by 95% and limited livestock rabies within a control area. However, effects were highly localized, with outbreaks continuing several kilometers from the culled area [28]. This suggests that culls might require very small populations of bats to reduce rabies risk, a target that could not be practically achievable in most areas. Consistent with this idea, spillover to livestock persists or has increased in some areas with regular culling, although the specific effects of culling have not been tested and it is conceivable that increases could reflect reductions in the amount of culling [20,47]. Detailed studies directly quantifying the impacts of culling on bat behavior, demography and rabies transmission in different environmental contexts are urgently needed.

### 3.3. Improving Management

For all non-bat reservoirs of rabies, vaccination of reservoirs is the cornerstone of human rabies prevention, having been applied successfully in both dogs and wild carnivores [48,49,50]. Bats have similar or longer lifespans (e.g., up to 15 years for wild common vampire bats [51]) than currently vaccinated carnivores (e.g., up to 12 years for wild red foxes [52]), but 3- to 6-fold slower reproduction, which would extend vaccine-induced population immunity to a point that effective herd immunity may prevent transmission definitively [53,54]. Moreover, naturally acquired VNAs from sublethal exposures potentially creates substantial baseline immunity not observed for carnivore reservoirs, which may also help reach effective herd immunity faster [55]. Existing recombinant viral vaccines using vaccinia [56,57] and raccoonpox [58,59] vectors are immunogenic and protective in bats. As both vectors are already used in large-scale campaigns targeting wildlife, they have been extensively tested for safety and lack of reversion to virulence in non-target species [48,60]. Furthermore, tools for mass dissemination in topical gels that enhance transfer between bats by allogrooming are being developed [58]. Applying this approach across many colonies remains a challenge, but vaccine releases might be optimized using knowledge of *D. rotundus* social, dispersal and reproductive behavior [61]. Alternatively, vaccines with greater potential to spread (i.e., “transmissible vaccines”) could be developed, ensuring that the potential negative effects of spreading a genetically-modified micro-organism in wild populations are minimized [62]. Field trials examining the dynamics of vaccine spread in the wild are a crucial next step. However, even if vaccination of bats minimized rabies circulation, the negative consequences of bat bites on humans and livestock would still demand improved strategies for bat population management.

## 4. Cross-Species Exposures

### 4.1. Current Knowledge

*D. rotundus* requires blood every 2–3 days to avoid starvation [63,64], and infectious bats can feed for days before succumbing to rabies. As such, bat bites are frequent and foraging behaviors define risks of cross-species transmission. When present, domestic livestock are fed on more frequently, but some individuals feed predominately on wildlife even when livestock are present [12,65]. Among livestock, cattle are the most attacked prey, but other species (e.g., horses, chickens, goats and pigs) can be the most common prey at specific locations depending on local prey diversity [18,66]. Bats forage over relatively small areas (typically <5 km) and bite frequency decreases with distance from roosts [67], which means that local prey availability in rural areas directly influences rabies risk to humans.

### 4.2. Current Management Practices

By reducing bat populations, culls also reduce bite rates (see above section). Bites on humans could be further reduced by “bat proofing” houses to prevent entry and by using mosquito nets to protect against bats within houses [68]. Although widespread implementation of these methods may be financially prohibitive in low-income communities and their efficiency has not been tested, netting might be beneficial for several mosquito-borne diseases and some costs to at risk-inhabitants might be absorbed by national programs aiming to reduce these diseases (e.g., malaria control in Brazil [69]). Use in rainforests where bites are most frequent may be further limited by the discomfort of closed wall houses and mosquito nets in hot, humid environments [68,70]. Human rabies outbreaks have followed the removal of livestock or depletion of wildlife prey [19,70], suggesting that introducing livestock might have a “zooprophylactic” effect of diverting bites from humans. However, this potential intervention must be considered with caution since supplementation of food resources may increase *D. rotundus* populations in the long term. Moreover, livestock may not be desirable in protected areas (i.e., national parks or protected forests) for environmental reasons such as deforestation for grazing. Preventing attacks on livestock might be achieved with physical barriers including fenced corrals analogous to mosquito nets, but high costs limit widespread implementation. Illuminating corrals may reduce attacks since bats avoid light but carries a high cost of electric power [71], which requires finding low-cost sources of electricity such as solar-powered lights. Furthermore, habituation of bats to lighted urban environments suggests that benefits might be short lived and potentially counterproductive for rabies control if deterrents promote bat dispersal [8].

### 4.3. Improving Management

Reducing bat bites is among the most challenging areas for successful intervention since at risk human populations are unlikely to adopt expensive new barriers or increase use of existing barriers that negatively impact daily well-being. Human exposures within houses could be reduced by identifying and changing factors limiting the use of mosquito nets and identifying populations at risk where bat-proof houses can be cost effective. Reducing bites in livestock is more challenging since livestock are mostly kept in unprotected corrals [18] and deterrents such as light and ultrasound [72] may have short-lived benefits or unintended consequences for bat dispersal. New approaches to cattle management may provide opportunities. Anecdotical evidence suggests that free-ranging cattle in large open pastures are less attacked by *D. rotundus* than those kept in corrals [73]. This suggests that, where possible, modifying farming practices (e.g., herd composition or distribution in space and time) could reduce bat foraging success. However, free-ranging cattle require considering other factors such as the need for concentrating cattle for routine vaccination and less protection from wild carnivore attacks (e.g., jaguars or mountain lions). Although new physical barriers against bat bites may eventually be developed, selective culling or reproductive suppression are presently the only foreseeable options to reduce cross-species exposures.

## 5. Susceptibility of Recipient Hosts

### 5.1. Current Knowledge

Rabies has the highest case fatality rate of any infectious disease, approaching 100% in untreated hosts, but for reasons that are still unclear, the likelihood of developing a productive infection which may be transmitted onward following exposure appears to vary by species [74]. Recent studies have demonstrated seroconversion in apparently healthy humans [68], livestock [75], and wildlife [76]. A wide range of vaccines confer protective immunity [77].

### 5.2. Current Management Practices

In humans, pre-exposure vaccines are available, but only imply a reduced schedule of prophylaxis after exposures. Pre-exposure vaccination is rarely provided to at-risk communities because it is costly and difficult to implement in remote areas where DRRV circulation is most problematic [2,68]. Community resistance to vaccination (e.g., local religious beliefs) is a separate impediment [78]. A tragic and unacceptable consequence is that vaccination campaigns are largely reactive to reports of human mortalities and mostly rely on post-exposure prophylaxis of previously bitten individuals. For livestock, deliberate post-exposure vaccination is not practiced, although some DRRV-incubating animals are likely to be incidentally vaccinated when vaccination occurs in response to an outbreak [2]. Preventive vaccines in livestock are effective but require annual boosters which inflate costs and reduce usage. In spite of this, benefits of preventive vaccination of livestock are over 6-fold higher than the costs associated with DRRV-associated mortality [2,79].

### 5.3. Improving Management

Socio-economical and anthropological research identifying and addressing barriers to widespread vaccination should be a priority for human and animal health. Transdisciplinary research has a vital role in shifting from reactive to preventive vaccination strategies. For example, spatiotemporal risk maps of DRRV distribution could prioritize areas for large-scale preventive vaccination of humans, while educational campaigns should aim to convince local populations of the benefits and safety of vaccines. Additional studies that characterize the true burden of rabies in humans and livestock should be undertaken to inform decisions on vaccine distribution [2,4]. Vaccines could also be improved by reducing costs and ensuring vaccines have a standard protective level across manufacturers. Recent progress in the route of administration (i.e., intradermal favored over intramuscular) has reduced the volume and number of doses for pre- and post-exposure prophylaxis of rabies recommended by the WHO [80]. Furthermore, developing vaccines that require fewer doses and that do not require a strict cold chain would benefit remote communities [2]. This should follow the recent progress made in dog rabies vaccination and the development of thermostable, single-dose vaccines [81].

## 6. Conclusions and Future Steps

A century has passed since rabies transmitted by *D. rotundus* was first reported in humans and livestock [6]. Despite decades of investments in bat culling, human vaccination, and animal vaccination, results throughout Latin America show that the problem is simultaneously increasing in at least three potentially interrelated ways: geographic expansions into historically rabies-free areas, increased incidence in endemic areas, and changes in the behavioral ecology of the reservoir host which put new species and localities at risk. As such, despite being the first discovered and among the best understood bat-borne zoonoses, rabies caused by DRRV can still reasonably be considered as an emerging public health threat. We argue that this revelation mandates a new generation of interventions that harness advances from fields spanning ecology, sociology, geography, computational biology, immunology and vaccinology to empower a shift from damage control to prevention. Crucial research that needs to be addressed now includes landscape analyses of bat distribution, quantification of the population-level impacts of existing and future vaccines targeting bats, as well as social scientific studies that identify the conditions under which existing and novel interventions will be accepted by high-risk populations (Figure 2). Technological developments, including vaccines and reproductive suppressants for bats and improved vaccines for livestock and humans, are within sight and could be transformative. Finally, there are substantial uncertainties in the scale of the problem and the effectiveness of interventions. Surveillance data are challenging to consolidate at all except the coarsest spatiotemporal scales (Figure 1). Under-reporting of mortality is likely widespread but, with few exceptions, is unquantified. Traditional control measures against DRRV such as culling have been applied for decades, but their efficacy is rarely quantified, or if quantified, is not publicly reported. We encourage the integration of international and interdisciplinary research teams to address these gaps and consolidate information to find emergent patterns that reveal context-dependent successes of different strategies. Collaborations between researchers and managers would create a mutually beneficial feedback, whereby successes and shortcomings of interventions would guide the development and implementation of the next generation of strategies to control and ultimately eliminate rabies transmitted by *D. rotundus*.

## Figures and Tables

**Figure 1 viruses-12-01002-f001:**
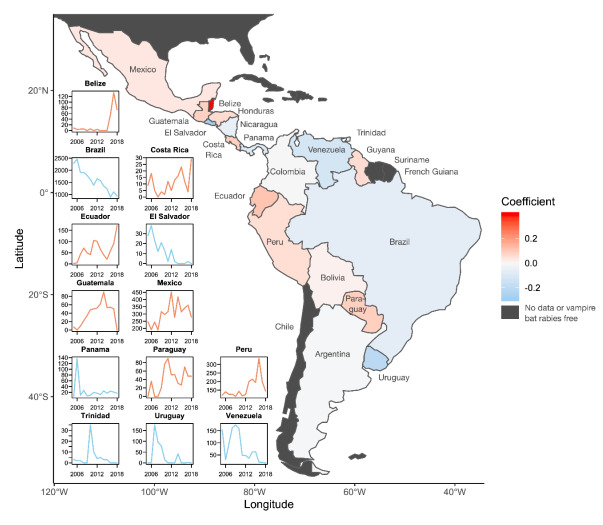
Geographic trends in rabies incidence across Latin America. Countries are colored according the coefficient estimated from a Poisson-distributed generalized linear model relating year to the number of reported rabies cases in livestock (*N* = 40287) and *Desmodus rotundus* (*N* = 214). Reds indicate increases and blues indicate decreases in rabies from 2005 to 2018. The inset panels show time series for countries with statistically significant trends (*p* < 0.05). Countries without DRRV or where data were unavailable are colored in dark grey. Data from the World Organization for Animal Health (OIE).

**Figure 2 viruses-12-01002-f002:**
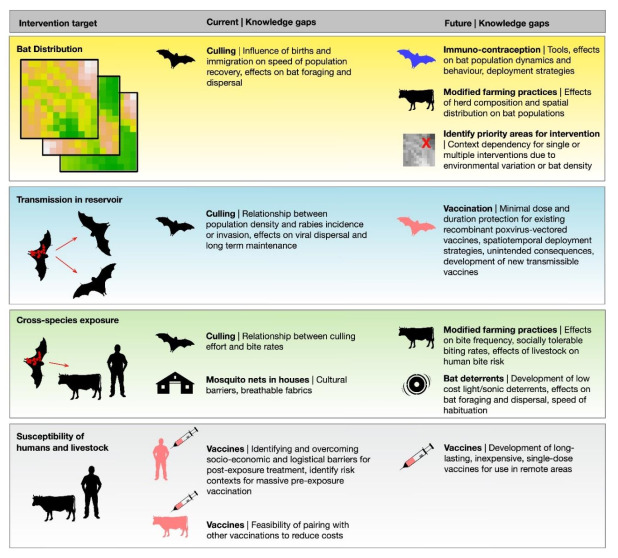
Filling research gaps to optimize current and future interventions against *Desmodus rotundus* rabies virus. All silhouettes were obtained from creazilla.com and are available on an open source license. Figures were made in R software.

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
