# Peer review of "Defining New Pathways to Manage the Ongoing Emergence of Bat Rabies in Latin America"

_viruses, 2020, doi:10.3390/v12091002_

Round 1

Reviewer 1 Report

Review “Defining New Pathways to Manage the Ongoing emergence of Bat Rabies in Latin America” by Benavides et al is a well written review by a number of experts on vampire bat rabies. This manuscript will be a valuable addition to the field after a few minor changes in the text.

Figure 1-This is an excellent illustration in the manuscript but is not very sharp.  Would recommend improving.

Line 134. Have ethical concerns been raised about this type of bat control?

Line 167.  Captive bat populations are likely to have increased stress levels.  This could affect the functioning of the immune system, altering the animals ability to control an infection it was exposed to prior to entering captivity.

Line 176.  Are localized disappearances confirmed and based on more than modeling data?   In the referenced manuscript the samples were collected from species in which spill over occurred as opposed bats or collecting sick acting bats at the roost.

Line 211.  Please give a numerical range for lifespan of bats and wild carnivores.

Line 215.  What about Gold et al, Plos Neglected tropical diseases 2020 https://doi.org/ 10.1371/journal.pntd.0007933

  1. Although not necessary, it may be nice to include a sentence that these nets may be beneficial against other vector borne diseases endemic in those areas. Additionally, aren’t there government programs to help defray the cost? Is it not enough or are these people too remote?

Line 254.  If lighting was helpful could solar power be used to decrease the cost?

Line 306 Please provide a citation.

Line 312 Rabies by definition is classified as an EID

https://www.niaid.nih.gov/research/emerging-infectious-diseases-pathogens

Author Response

Review “Defining New Pathways to Manage the Ongoing emergence of Bat Rabies in Latin America” by Benavides et al is a well written review by a number of experts on vampire bat rabies. This manuscript will be a valuable addition to the field after a few minor changes in the text.

We thank the reviewer for their useful comments, and we have addressed each one of them bellow.

Figure 1-This is an excellent illustration in the manuscript but is not very sharp.  Would recommend improving.

We have now increased figure resolution to 750 dots per inch to make it sharper.

Line 134. Have ethical concerns been raised about this type of bat control?

Yes, we added a statement and references on the ethical concerns of this type of bat control (Lines 135-137)

Line 167.  Captive bat populations are likely to have increased stress levels.  This could affect the functioning of the immune system, altering the animals ability to control an infection it was exposed to prior to entering captivity.

We agree with the reviewer and have included this possibility in the sentence (Lines 171-172)

Line 176.  Are localized disappearances confirmed and based on more than modeling data?   In the referenced manuscript the samples were collected from species in which spill over occurred as opposed bats or collecting sick acting bats at the roost.

To our knowledge, no study has definitively shown localized disappearances from bats themselves, but this would require longitudinal sampling which is rarely carried out. However, available evidence for viral extinctions is not purely based on modeling, but rather reflects long term absences of outbreaks in livestock and phylogenetic evidence of viral lineage replacement.  Nonetheless, we agree that confirming this pattern in bats would be valuable and we highlight the need for longitudinal studies using RT-PCR to accomplish this (Lines 181-186).

Line 211.  Please give a numerical range for lifespan of bats and wild carnivores.

Numerical range has been provided (up to 15 years for wild vampire bats and up to 12 years for the red fox) as well as references supporting this statement (Lines 220-221)

Line 215.  What about Gold et al, Plos Neglected tropical diseases 2020 https://doi.org/ 10.1371/journal.pntd.0007933

We thank the reviewer for this reference. Added (Line 225)

Line 243. Although not necessary, it may be nice to include a sentence that these nets may be beneficial against other vector borne diseases endemic in those areas. Additionally, aren’t there government programs to help defray the cost? Is it not enough or are these people too remote?

We have added a sentence acknowledging the benefits of these nets for other mosquito-borne diseases and added a sentence stating how national programs (e.g. malaria control Brazil) can deliver free-of-charge mosquito nets (Lines 254-256).

Line 254.  If lighting was helpful could solar power be used to decrease the cost?

We have added the necessity of finding low-cost power sources such as solar power to decrease cost (Lines 266-267).

Line 306 Please provide a citation.

The sentence was modified to include both humans and livestock, and a reference was provided (Lines 319-320)

Line 312 Rabies by definition is classified as an EID

https://www.niaid.nih.gov/research/emerging-infectious-diseases-pathogens

We agree that rabies can be classified as EID or re-emerging  (https://www.ncbi.nlm.nih.gov/pmc/articles/PMC3322764/#R2), but think that this point is sometimes neglected in conversations focused on other emerging bat viruses, so think it is valuable to emphasize. We have therefore modified this sentence to include the term ‘emerging public health threat (Line 326). We also updated the term in the abstract.

Reviewer 2 Report

This review article overviews knowledge on epidemiology of rabies in D. rotundus (bat rabies) and its management in Latin America, explicating the status and problems of current interventions against bat rabies targeting i) bat distribution, ii) transmission in reservoir, iii) cross-species exposure and iv) susceptibility of humans and livestock. This paper also discusses what/how the management of bat rabies should be in the future, considering the current problems.

This is a well-written, very informative article, providing useful information on bat rabies management in Latin America. As a reviewer, I appreciate quality of this paper.

I have only a few comments as follows:

  • The resolution of Figure 1 should be improved. Texts look blurry especially after printing out.
  • In some paragraphs such as 2.2 and 2.3, “D. rotundus” are not shown in Italic.
  • there efficiency has not been tested (line 243): “there”might be a mistake for “their”or “therefore”?

Author Response

This review article overviews knowledge on epidemiology of rabies in D. rotundus (bat rabies) and its management in Latin America, explicating the status and problems of current interventions against bat rabies targeting i) bat distribution, ii) transmission in reservoir, iii) cross-species exposure and iv) susceptibility of humans and livestock. This paper also discusses what/how the management of bat rabies should be in the future, considering the current problems.

This is a well-written, very informative article, providing useful information on bat rabies management in Latin America. As a reviewer, I appreciate quality of this paper.

We thank the reviewer for their useful comments, and we have addressed each one of them bellow.

I have only a few comments as follows:

  • The resolution of Figure 1 should be improved. Texts look blurry especially after printing out.

The resolution of the figure has been increased.

  • In some paragraphs such as 2.2 and 2.3, “D. rotundus” are not shown in Italic.

Italic has been used across the manuscript.

  • there efficiency has not been tested (line 243): “there”might be a mistake for “their”or “therefore”?

Changed to ‘their’ (Line 253).

Round 2

Reviewer 1 Report

Edits have been made to the manuscript improving clarity and can be published in current form

Author Response

We thank the review for the comments. We have now addressed them as follow:

- Line 292: I suggest to update the reference 5 with the last (third) version of the WHO report (2018).   We have updated the reference (line 292)   - Lines 314-315: Recent progress has been made for human pre- and post-exposure prophylaxis of rabies (route of administation with lower volume of vaccine, reduction of number of doses). I suggest to the authors to mention it, with ad hoc references (for example Rabies vaccines: WHO position paper – April 2018).   We have added this information including the following statement:   'Recent progress in the route of administration (i.e. intradermal favored over intramuscular) have reduced the volume and number of doses for pre- and post-exposure prophylaxis of rabies recommended by the WHO [82]'. (Lines 315-317)